# New Incretin Combination Treatments under Investigation in Obesity and Metabolism: A Systematic Review

**DOI:** 10.3390/ph14090869

**Published:** 2021-08-28

**Authors:** Agni Kakouri, Georgia Kanti, Efthymios Kapantais, Alexandros Kokkinos, Leonidas Lanaras, Paul Farajian, Christos Galanakis, Georgios Georgantopoulos, Nikos F. Vlahos, George Mastorakos, Alexandra Bargiota, Georgios Valsamakis

**Affiliations:** 1Athens Medical School, National and Kapodistrian University of Athens, 115 27 Athens, Greece; agni.kakouri@gmail.com; 2Department of Ophthalmology & Visual Sciences, University of Illinois, Chicago, IL 60607, USA; 3Endocrinology and Diabetes Center, Athens General Hospital “G. Gennimatas”, 115 27 Athens, Greece; g.kanti89@gmail.com; 4Hellenic Medical Association for Obesity, 115 27 Athens, Greece; ek@kapantais.gr (E.K.); rjd@otenet.gr (A.K.); leolanaras@gmail.com (L.L.); pfarajian@nutrimed.gr (P.F.); christos@myfamilydoctor.gr (C.G.); gg.psych.mail@gmail.com (G.G.); 5Assisted Reproduction Unit, 2nd Department of Obstetrics and Gynecology, Aretaieion Hospital, Medical School, National and Kapodistrian University of Athens, 115 28 Athens, Greece; nfvlahos@gmail.com; 6Department of Endocrinology, Diabetes Mellitus and Metabolism, 2nd Department of Obstetrics and Gynecology, Aretaieion University Hospital, 115 28 Athens, Greece; mastorakg@gmail.com; 7University Department of Endocrinology and Metabolic Disorders, University Hospital of Larissa, University of Thessaly, 413 34 Larissa, Greece; abargio@uth.gr

**Keywords:** obesity, metabolism, anti-obesity combined medications, weight loss, incretins

## Abstract

The worldwide upward trend in obesity in adults and the increased incidence of overweight children suggests that the future risk of obesity-related illnesses will be increased. The existing anti-obesity drugs act either in the central nervous system (CNS) or in the peripheral tissues, controlling the appetite and metabolism. However, weight regain is a common homeostatic response; current anti-obesity medications show limited effectiveness in achieving long-term weight loss maintenance; in addition to being linked to various side effects. Combined anti-obesity medications (per os or injectable) target more than one of the molecular pathways involved in weight regulation, as well as structures in the CNS. In this systematic review, we conducted a search of PubMed and The ClinicalTrials.gov up to February 2021. We summarized the Food and Drug Administration (FDA)-approved medications, and we focused on the combined pharmacological treatments, related to the incretin hormones, currently in a clinical trial phase. We also assessed the mechanism of action and therapeutic utility of these novel hybrid peptides and potential interactions with other regulatory hormones that may have beneficial effects on obesity. As we improve our understanding of the pathophysiology of obesity, we hope to identify more novel treatment strategies.

## 1. Introduction

Obesity is a chronic metabolic disease, which can be defined as an excessive accumulation of body fat that leads to increased morbidity and mortality. Obesity has been shown to increase the risk of various medical conditions, such as diabetes mellitus type 2, cardiovascular disease, gallbladder disease, nonalcoholic fatty liver disease, several types of malignancies (i.e., in colon, endometrium, breast, esophagus, kidney), osteoarthritis, obstructive sleep apnea, and mental health problems, leading to increased morbidity and mortality [1]. The worldwide prevalence of obesity has nearly tripled between 1975 and 2016 and continues to rise at an alarming rate, rendering it a global pandemic [2]. It is further estimated that up to 58% of the world’s adult population could be overweight or obese by 2030 if the upward trends continue [3]. Obesity in adults is defined as a BMI of 30 kg/m^2^ and above and overweight as a BMI between 25 and 29.9 kg/m^2^ [4]. The pathophysiology of the disease is complex, and multiple factors are involved. Obesity arises from a chronic positive energy balance, which occurs when energy intake exceeds energy expenditure [5]. A better understanding of the body weight regulation mechanisms and human metabolism remains challenging and will help create effective prevention and treatment strategies against obesity. Currently approved anti-obesity medications have variable weight loss response and weight maintenance along with essential adverse effects, which render it unavoidable to discontinue the therapy [6]. Therefore, it is of great importance to explore new anti-obesity agents with combined actions on multiple levels in the pathophysiological mechanisms involved in the progression and maintenance of obesity.

## 2. Methods

A literature search was conducted using the database of PubMed and The ClinicalTrials.gov from September 2020 up to February 2021. A query of clinical trials registries (Clinicaltrials.gov) with the keyword “obesity” returned 9532 search results, which further contained 769 interventional completed studies with associated results using the keyword “interventional studies”. Out of the 769 studies, 492 were evaluating the effect of drugs either on obesity or in obese subjects. Out of 492, we evaluated only 56 clinical trials that provided available published results and were related to anti-obesity medications and incretin hormones.

## 3. Results

### 3.1. Approved Pharmacotherapy for Obesity

Over the last decade, five anti-obesity medications have received approval by the US Food & Drug Administration (FDA), providing promising anti-obesity therapeutic options (Table 1): orlistat, phentermine/topiramate, bupropion/naltrexone, liraglutide, and more recently, semaglutide [7,8,9,10]. The European Medicinal Agency (EMA) has approved only four of them: orlistat, bupropion/naltrexone, liraglutide, and semaglutide. In February 2020, Lorcaserin was withdrawn from the US market, as clinical trials showed an increased occurrence of malignancies [11]. Most of the FDA-approved anti-obesity medications are administered orally, except for liraglutide and semaglutide, which require daily injectable subcutaneous administration [12,13]. The existing anti-obesity drugs act either in the central nervous system (CNS) or in the peripheral tissues, controlling appetite and metabolism. The currently approved pharmacotherapies achieve a weight reduction of 5–15% [14].

### 3.2. Orlistat

Orlistat acts by inhibiting pancreatic lipases, resulting in a reduction of intestinal fat digestion. It has been proved to be efficient in enabling weight loss, which was maintained up to 24 to 36 months after treatment [21]. In a 24-week double-blinded, placebo-controlled study, 120 mg of orlistat was administered three times daily. Bodyweight in the orlistat group was significantly reduced (6.8% from baseline) compared to the placebo group (3.8%). Furthermore, 64% of the subjects in the orlistat group achieved 5% weight loss in comparison with 39% in the placebo group [16]. Additionally, it has been shown to improve metabolic parameters, including the incidence of diabetes [22], blood pressure [23], and serum lipid levels [24]. The main adverse effects of orlistat are diarrhea, abdominal pain, and steatorrhea [25], while severe liver [26] and acute renal injury [27] have been reported only in a few cases. It may also lower the levels of fat-soluble vitamins [28]. The main contraindications reported are related to pregnancy, malabsorption syndrome, and cholestasis [29]. The drug is provided in 120 mg capsules and is recommended to be taken three times daily.

### 3.3. Phentermine/Topiramate

This combination drug aims at a greater efficacy with lower toxicity, as the two substances act synergistically. Phentermine is a sympathomimetic amine, which stimulates norepinephrine release, resulting in reduced appetite. Topiramate is an anticonvulsant, which was shown to have an anorexigenic action, thus inducing weight loss. The combination drug phentermine/topiramate was approved by the FDA in 2012 in a recommended dose of 7.5 mg phentermine/46 mg topiramate once daily [30]. The approval was based on three clinical trials: EQUIP [31], CONQUER [32], and SEQUEL [19]. In the CONQUER study, 75% of participants exhibited a reduction in body weight of >5% of baseline, as well as improvement in blood pressure, lipid levels, fasting glucose, and a lower waist circumference. In these trials, the most common adverse reactions reported were constipation, paresthesia, and dry mouth. The combination drug is contraindicated in pregnancy and uncontrolled hypertension [33].

### 3.4. Bupropion/Naltrexone

Bupropion/naltrexone is a combination drug, which consists of two already approved centrally acting medications. It acts by reducing appetite through the activation of POMC neurons in the arcuate nucleus. Bupropion is a dopamine and norepinephrine-reuptake inhibitor, used as an antidepressant and for smoking cessation [34]. Naltrexone is an opioid receptor antagonist, prescribed for alcohol or opiate dependence [35]. The combination of bupropion/naltrexone was approved by the FDA in 2014 [36]. It has been shown to reduce weight by approximately 4–5% compared to placebo [37]. In a 56-week placebo-controlled trial, weight loss was 9.3 ± 0.4% in the bupropion/naltrexone group compared to 5.1 ± 0.6% in the placebo group. Significantly more participants (66.4%) in the bupropion/naltrexone group lost >5% of their body weight at week 56 compared to 42.5% in the control group [18]. Regarding the adverse effects, nausea, headache, and constipation were most frequently reported [38]. Pregnancy, seizures, anorexia/bulimia, alcohol withdrawal, and/or the use of monoamine oxidase inhibitors within 2 weeks of drug initiation are the major reported contraindications [39]. The recommended dose is 16 mg naltrexone/180 mg bupropion twice daily.

### 3.5. Liraglutide

Liraglutide, a glucagon-like peptide-1 (GLP-1) receptor agonist, is administered subcutaneously (3 mg once daily) [40]. GLP-1 agonists activate GLP-1 receptors in the central nervous system. They stimulate the POMC/CART neurons, which have anorexigenic action, and they inhibit orexigenic neurons, which express neuropeptide Y [41]. Moreover, they improve glucose-dependent insulin release, suppress glucagon release, and decrease the rate of gastric emptying [42,43]. In the Satiety and Clinical Adiposity-Liraglutide Evidence in non-diabetic and diabetic people (SCALE) study, subjects were enrolled in a 56-week, randomized, double-blind, placebo-controlled, multicenter trial in which individuals with obesity received liraglutide 3.0 mg or placebo in combination with intensive behavioral therapy. At 56 weeks, the mean weight loss in the placebo group was 4.0%, while in the treatment group, it was 7.5%. More than 5% weight loss was achieved in 61.5% of subjects in the liraglutide group compared to 38.8% in placebo [44]. In the XENSOR study, weight reduction was 7.7 kg, significantly greater compared to 3.3 kg with orlistat. A total of 64.7% of patients lost >5% of their baseline weight compared with orlistat, of which only 27.4% achieved this endpoint [15]. The most common side effect associated with GLP-1 therapy is short-term nausea. Other common adverse effects include hypoglycemia, diarrhea, constipation, vomiting, headache, decreased appetite, dyspepsia, fatigue, dizziness, abdominal pain, and increased levels of lipase [45]. The main contraindications reported are related to pregnancy, multiple endocrine neoplasia 2 syndrome, history of medullary thyroid cancer and acute or chronic pancreatitis [46]. Attenuating hyperglycemia during meals is particularly beneficial in patients with hyperglycemia or diagnosed diabetes [47].

### 3.6. Semaglutide

Semaglutide is a long-acting GLP-1 receptor agonist. Semaglutide mimics the effects of GLP-1 by increasing satiety, reducing appetite for food, prospective food consumption, and energy intake [41]. Semaglutide decreases blood glucose by increasing insulin secretion and decreasing glucagon secretion, both in a glucose-dependent manner [12]. A 68-week randomized, double-blind, placebo-controlled trial investigated the effects of once-weekly subcutaneous semaglutide injection, at a dose of 2.4 mg, in 1961 obese patients. Semaglutide resulted in a 14.9% mean weight reduction from baseline compared with a 2.4% reduction in the placebo group (95% CI −13.4 to −11.5; *p* < 0.001). A total of 86.4% of participants in the treatment group achieved weight reductions of 5% or more compared to 31.5% in the placebo group. Participants in the semaglutide group had improved cardiometabolic risk factors and fewer food cravings. The most frequent side effects were dose-related nausea and diarrhea [17]. Semaglutide should not be used in patients with history of medullary thyroid carcinoma or multiple endocrine neoplasia syndrome type 2, as it increases the risk for thyroid c-cell carcinoma [48]. Contraindications are similar to liraglutide and the rest of the GLP-1 agonists since they have equal mechanisms of action. In another randomized, double-blind, placebo-controlled, multicenter phase 2 clinical trial, 957 individuals were enrolled. The mean weight loss for the placebo group was −2.3% compared to −6% (0.05 mg), −8.6% (0.1 mg), −11.6% (0.2 mg), −11.2% (0.3 mg), and −13.8% (0.4 mg) for the semaglutide subcutaneous groups. Weight loss of more than 10% occurred in 37–65% of participants receiving 0.1 mg or more of semaglutide compared to 10% in the placebo group (*p* < 0.001). Furthermore, the liraglutide group (3.0 mg subcutaneous) had a 7.8% reduction in body weight significantly less than the 11–14% reduction with semaglutide doses of 0.2 mg or more. Consistent improvements in glucose and cardiac parameters such as lipids and high-sensitivity C-reactive protein were seen for the semaglutide group versus placebo [49]. The PIONEER trial program is evaluating the efficacy and safety of oral semaglutide in diabetes type II. Patients were randomly assigned to oral semaglutide 3 mg, 7 mg, 14 mg, or placebo. All doses of oral semaglutiide resulted in a reduction in HbA1C. The 14 mg semaglutide provided statistically significant higher reductions in body weight compared to placebo at 26 weeks. In another part of the PIONEER trial, oral semaglutide (dose-escalated to 14 mg) was compared to liraglutide (dose-escalated to 1.8 mg) for 52 weeks. The results showed that HbA_1C_ was significantly decreased in the semaglutide group compared to the liraglutide group. Similarly, weight loss was increased in the semaglutide group compared to the liraglutide with an estimated treatment difference of −1.3 kg (*p* < 0.001) and −3.3 kg for the placebo group. Consequently, oral semaglutide may be a breakthrough treatment for patients with diabetes type II or obesity who are not inclined to have an injectable therapy [50,51,52,53,54,55].

### 3.7. Pharmacotherapy for Obesity under Investigation

Recent research has suggested that the therapeutic implications of incretin hormones as combination agents in obesity, such as GLP-1 plus GIP, GLP-1 plus glucagon, GLP-1 plus GIP plus glucagon receptor agonists, GLP-1 plus amylin analogue, may have promising potential for weight loss and weight maintenance.

#### Biology of Incretins

Incretins are peptide hormones secreted by the gastrointestinal tract after nutrient intake. The two major incretins are glucagon-like peptide-1 (GLP-1) and glucose-dependent insulinotropic polypeptide or gastric inhibitory polypeptide (GIP) [56]. GLP-1 is a 30-amino acid peptide hormone, secreted from the intestinal endocrine L-cells, mainly located in the ileum but also in the duodenum, colon, and rectum. This hormone is “meal-related”, meaning that during the fasting state, plasma concentrations are very low, while during meals, especially those rich in fats and carbohydrates, GLP-1 secretion is rapidly enhanced [57]. GIP is a 42-amino-acid peptide which is secreted from K-cells; specific endocrine cells located on the duodenum and proximal jejunum [58]. GIP is secreted in response to the ingestion of nutrients such as fat and carbohydrates [40]. Dipeptidyl peptidase 4 (DPP-4) enzyme is responsible for the rapid degradation of GLP-1 and GIP to inactive metabolites. DDP-4 is a serine protease that specifically cleaves dipeptides from the amino terminus of proteins that contain an alanine or proline residue in the second position [59]. The actions of GIP and GLP-1 are mediated by the engagement of G-protein-coupled receptors expressed on pancreatic α- and β-cells and in various tissues, including the gastrointestinal tract, kidneys, central nervous system, heart, and lungs. Stimulation of the incretin receptors activates adenyl cyclase, which increases cyclic adenosine monophosphate c-AMP levels and protein kinase A activation. This leads to insulin secretion in a glucose-dependent manner [60]. GLP-1 results in enhanced insulin secretion [42] and GIP stimulates insulin secretion by β-pancreatic cells in a glucose-dependent manner [43]. GLP-1 suppresses glucagon secretion from pancreatic a-cells only during states of hyperglycemia and euglycemia which reduces hepatic glucose production. During hypoglycemia, glucagon release is fully preserved, even in the presence of pharmacological concentrations of GLP-1. Interestingly, insulinotropic but not glucagon static effects of GLP-1 are significantly reduced in patients with diabetes [57]. On the contrary, GIP enhances postprandial glucagon response, stimulates glucagon secretion from pancreatic a-cells during a state of hypoglycemia or euglycemia but not during hyperglycemia and stimulates glucagon secretion. The fundamental effect associated with these hormones is called the incretin effect. Oral glucose leads to amplified insulin secretion by the pancreatic β-cells compared to the intravenous glucose administration, despite inducing similar levels of glycemia, in healthy individuals [56]. This phenomenon is explained by the fact that oral glucose leads to the release of incretin hormones from the gastrointestinal tract, while intravenous glucose does not. Furthermore, GLP-1 reduces gastrointestinal secretion by exhibiting an inhibitory effect on pentagastrin and meal-stimulated gastric acid secretion [43]. It also diminishes gastrointestinal motility, which contributes to the normalization of blood glucose levels in type II diabetes mellitus after exogenous GLP-1 administration and promotes satiety by ensuing reduction of food intake [44]. Additionally, GLP-1 has a direct effect on hypothalamic neurons and regulates energy homeostasis and food consumption [43]. GIP has been associated with increased blood flow to adipose tissue and intestines as well as with enhanced triacyl glyceride deposition in the adipose. The anabolic effects of GIP include stimulation of fatty acid synthesis, re-esterification, incorporation of fatty acids into triglycerides stimulated by insulin, lipoprotein lipase synthesis, and lowering of glucagon-stimulated lipolysis [56].

### 3.8. GIP/GLP-1 Receptor Agonists

A dual non-balanced agonist of GIP receptor and GLP-1 receptor, tirzepatide (also known as LY3298176), has been developed for a once-weekly injectable administration [61]. The combination of insulinotropic actions of GIP with the favorable cardiovascular outcomes and weight reduction of GLP-1 may offer superior glycemic control in the treatment of diabetes mellitus type II [62]. Compared to a selective GLP-1 receptor agonist, the synergistic action of GIP combined with GLP-1 shows more effective weight loss and reduction in food intake with increased energy expenditure in genetic mouse models with diet-induced obesity [61]. Increasing doses of tirzepatide (2.5–15 mg once weekly subcutaneous) over a period of 12 weeks was compared to placebo in 111 patients. Tirzepatide dose-dependently significantly reduced HbA_1C_. Tirzepatide also reduced weight by 5.7 kg from baseline in subjects receiving 15 mg compared to 0.5 kg in the placebo group [62]. In another 26-week phase 2b clinical trial, 316 subjects were enrolled and assigned randomly in three groups: the tirzepatide group, the dulaglutide group (a GLP-1 receptor agonist), and the placebo group. Tirzepatide, like dulaglutide, reduces fasting, postprandial plasma glucose and HbA_1C_. Bodyweight was reduced with tirzepatide (−0.9, −4.8, −8.7, and −11.3 kg for 1, 5, 10, and 15 mg, respectively) compared to a 2.7 kg reduction with dulaglutide and 0.4 kg weight loss in the placebo group. A total of 14–71% of subjects treated with tirzepatide achieved at least 5% weight reduction and 6–39% achieved the target of at least 10% weight reduction [63]. The most frequent side effects reported with LY3298176 were dose-dependent gastrointestinal symptoms, such as vomiting, nausea, decreased appetite, diarrhea, and abdominal distension [62,64]. The SURPASS phase 3 trials aim to assess the safety and efficacy of escalating doses of tirzepatide starting from 2.5 mg weekly until the maintenance dose of 15 mg [65]. The SURPASS-2 40-week trial compared increasing doses of tirzepatide (once weekly) versus semaglutide injections in 1879 patients. A total of 86% of the patients achieved HbA_1c_ reduction below 7% compared to 79% treated with semaglutide. Moreover, the average weight loss with tirzepatide was 1.9 kg and 5.5 kg, additional to what semaglutide achieved, for the 5 mg and 15 mg, respectively [66]. In clinicaltrials.gov, trials utilizing these dual agonists are still ongoing; however, the results of the studies are unpublished at the time of writing.

### 3.9. GLP-1/Glucagon Receptor Agonists

Oxyntomodulin is a 37-amino acid peptide hormone produced by enteroendocrine L-cells of the gastrointestinal tract after enzymatic processing [67]. Oxyntomodulin is released post-prandially in response to nutrients, such as carbohydrates, lipids, and proteins. Oxyntomodulin is subject to proteolytic cleavage by dipeptidyl-peptidase-4 and fast renal clearance. Oxyntomodulin’s main role is to suppress appetite by acting as an agonist of both GLP-1 and glucagon receptors and improve glucose metabolism [68]. The combination of lipolytic and thermogenic actions of glucagon with the anorectic and insulinotropic actions of GLP-1 is crucial for the targeted management of obesity [69]. Oxyntomodulin reduces food intake and enhances energy expenditure by activating both GLP-1 receptors and glucagon receptors. In a 4-week study, obese and overweight subjects were given oxyntomodulin subcutaneous injection (400 nmol) or placebo three times daily 30 min pre-prandial. At the end of the study, it was reported that subjects in the treatment group had an average weight loss of 2.4 ± 0.4 kg compared to the control group, which had an average weight reduction of 0.5 ± 0.6 kg. Moreover, energy intake was lower on day 2 and day 29 compared to day 1, suggesting that weight loss could be maintained over time [70]. Consequently, clinical experience with oxyntomodulin showed promising evidence that a three-times daily subcutaneous administration in 4 weeks provides vigorous weight loss to obese individuals [71]. Synthetic oxyntomodulin with optimal balancing of GLP-1 and glucagon receptor activities produces a significant and sustained increase in energy expenditure, compared to the gut hormone, as discussed by Scott. et al. [72]. Oxyntomodulin has been reported to enhance insulin secretion in short-term clinical studies in obese people with or without diabetes mellitus type II [73]. Another 6-week experimental study on diet-induced obese diabetic monkeys showed that a combination of GLP-1 receptor agonist and glucagon receptor agonist significantly reduced body weight by 6.6 ± 0.9%. However, glucagon receptor agonist alone had no effect on body weight while GLP-1 receptor agonist lowered body weight by 3.8 ± 0.9% and energy intake by 18% [74]. Few incidences of acute pancreatitis have been reported during treatment with GLP-1R agonists apart from gastrointestinal side effects [75]. Moreover, glucagon and GLP-1 have positive inotropic and chronotropic action on the heart with mild decreases in LDL-C and triglycerides levels [76,77]. Recent clinical studies showed that cotadutide, a dual receptor GLP-1/Glucagon agonist, improves glycemic control and promotes weight reduction. More specifically, in a 49-day placebo-controlled study, 65 obese subjects were given once-daily subcutaneous cotadutide. A total of 42% of subjects achieved an over 5% reduction in body weight compared to 8% in the control group. The bodyweight reduction was 3.41% from baseline in the cotadutide group versus 0.08% in the control group, suggesting promising results for the treatment of obesity [78].

### 3.10. GLP-1 Receptor Agonist/Sodium Glucose Co-Transporter-2 (SGLT2) Inhibitor

Under physiologic conditions, up to 180 g of glucose is filtered by the renal glomerulus daily and all of it is subsequently reabsorbed in the proximal convoluted tubule. This tubular glucose reabsorption is affected by the combined action of two sodium-dependent glucose cotransporter (SGLT) proteins. Once glucose has been reabsorbed into the tubular epithelial cells, it diffuses into the interstitial space across specific facilitative glucose transporters (GLUTs) [79]. GLT2 inhibitors block the pathway of SGLT2 glucose reabsorption in the proximal convoluted tubule, resulting in profound glucosuria and natriuresis [80]. SGLT2 inhibitors are used in the treatment of type II diabetes, while they exhibit cardioprotective and anti-atherosclerotic effects [80,81]. The combined action of SGLT2 inhibitors to cause caloric loss via glucosuria, with the GLP-1 receptor agonist to reduce appetite and caloric intake, presumably leads to additive effects on weight reduction. Moreover, GLP-1 receptor agonists and SGLT2 reduce total body weight by lowering the visceral fat via increasing urinary glucose excretion and subcutaneous adipose tissue volume [82]. Consequently, SGLT2 inhibitors are successful for the treatment of diabetes mellitus type II, and they also seem to reduce total body weight. In a randomized, double-blind placebo-controlled trial, 126 obese patients were randomly assigned to the placebo group or licogliflozin group, a dual SGLT1/2 inhibitor (LIK066), once daily. At the endpoint of 12 weeks, the percentage change in body weight from baseline was −1.86, −2.84, −3.41, and − 3.80% in licogliflozin 2.5, 10, 25, and 50 mg dose groups, respectively, compared to the control group, which was 0.11. The proportion of responders with a ≥3% reduction in body weight in the licogliflozin 2.5, 10, 25, and 50 mg dose groups were 15.8%, 55.6%, 50.0%, and 56.7%, respectively, versus placebo 7.1%; *p* ≤ 0.002 for all except the 2.5 mg once-daily group (*p* = 0.39). Licogliflozin also promoted a dose-dependent reduction in uric acid levels and HbA_1c_. Licogliflozin was well tolerated, and no safety issues raised [83]. A recent clinical trial studies the effects of combined dapagliflozin with GLP-1 analogue exenatide. The results demonstrated a mean weight loss of 4.5 kg over a period of 24 weeks that increased to 5.7 kg [84] over a period of 52 weeks, suggesting promising effects on sustained reductions in body weight [85] Another 28-week clinical trial evaluated the combination of exenatide plus dapagliflozin once daily versus exenatide or dapagliflozin alone. The dual combination led to significant weight loss, HbA_1C_, and blood pressure reduction with no unexpected side effects. The most usual adverse effects reported with dapagliflozin in clinical trials were female genital mycotic infections, urinary tract infections, and nasopharyngitis [86]. The weight change was −3.55 kg in the combination group versus −1.56 in the exenatide group and −2.22 in the dapagliflozin group. Weight loss of 5% or more was achieved in 33% of the dual combination group compared to 14% in the exenatide and 20% in the dapagliflozin group [87]. The safety of canagliflozin, when used in combination with a GLP-1 agonist, was evaluated in the CANVAS study in patients with diabetes type II. The most common adverse events reported were diarrhea, injection-site erythema, nausea, urinary tract infections, genital mycotic infections, and osmotic diuresis-related effects [88].

### 3.11. GLP-1 Receptor Agonist/Amylin Analogue

Cagrilintide is a novel long-acting acylated amylin analogue that is combined with semaglutide and leads to greater weight loss compared to semaglutide monotherapy [89]. Amylin is a 37-amino acid polypeptide that is co-released with insulin from β-pancreatic cells and is involved in delaying gastric emptying [90]. Cagrilintide has agonistic properties on amylin and calcitonin receptors. Recent evidence shows that amylin is expressed in certain central nervous system pathways that are involved in metabolism and energy regulation, such as the hypothalamus [91]. In a recent randomized, placebo-controlled 20-week study, a once-weekly subcutaneous dose of cagrilintide 4.5 mg therapy combined with subcutaneous semaglutide 2.4 mg led to 15.4% of weight loss compared with matched placebo that was 8%, without any lifestyle modifications. A total of 96 individuals were enrolled in the study and the results showed a 15.7% bodyweight reduction from baseline with 1.2 mg cagrilintide and 17.1% reduction with 2.4 mg cagrilintide compared to placebo (both in combination with semaglutide 2.4 mg). The most common side effect was gastrointestinal disorder, with nausea being the most frequently reported [92].

### 3.12. GLP-1/GIP/Glucagon Receptor Agonists

This new triagonist has been shown to decrease plasma glucose levels and to promote weight loss in obese rodents with higher metabolic benefits compared to diagonists such as GLP-1 and glucagon. Glucagon increases energy expenditure, and when combined with GLP-1, which acts as an appetite suppressant, results in a net loss of body weight. GIP receptor agonism and GLP-1 receptor agonism limit the hyperglycemia caused by glucagon and improves insulin sensitivity [42]. Consequently, the triagonist may improve the overall weight loss and metabolic profile of the patients. In an animal study by Hofmann S. et al., it was demonstrated that male and female mice with the same body fat mass had lower body weight, fat mass, and diminished food intake after being treated with injectable GLP-1/GIP/glucagon triple agonist. In addition, dyslipidemia was also improved in both female and male mice and diet-induced steatohepatitis was reversed in the female mice [93]. Another study in rodent models further showed that the triagonist considerably improved glucose tolerance and insulin sensitivity, and lowered food intake [94]. It also presented that mice had reduced body weight by 26.6% compared to GLP-1/GIP receptor agonist, which decreased weight by 15.7% after 20 days of treatment. Furthermore, the triagonist was reported to lessen the plasma cholesterol concentration to a higher extent compared with the GLP-1/GIP dual agonist. In a more recent study, Kannt A. et al. showed that mice that were placed on a specific diet, which led to non-alcoholic steatohepatitis and liver fibrosis, for 36 weeks lost 8% to 9% of their diet-induced body weight after 8 weeks of treatment with the combination of glucagon, GLP-1, and GIP. The main reason for the weight loss was the initial decrease in food consumption during the first week of the trial, which remained stable across the study period. Moreover, the triagonist led to an improvement in steatohepatitis, which was confirmed by a biopsy, of the diet-induced obese mice [95]. Based on the promising results of the animal studies, few GLP-1/GIP/Glucagon receptor agonists are currently under phase1 clinical trial investigation. These are the Hanmi Pharmaceuticals with the HM15211 compound and the Novo Nordisk/Marcadia with MAR423. Results regarding their safety and efficacy remain to be published.

### 3.13. GLP-1 Receptor Agonist/Oxyntomodulin/Peptide YY

GLP-1, peptide YY, and oxyntomodulin are all secreted by the L-cells in the small intestine and colon. Peptide YY (PYY) is a 36-amino acid gastrointestinal hormone which exhibits an anorexic effect via the neuropeptide Y, Y2 receptors, in the arcuate nucleus [86]. Studies showed that infusion of the PYY in healthy volunteers induces a 33% reduction in caloric intake over 24 h [96]. Other roles of PYY include delaying gastric emptying, altering colonic motility, and reducing postprandial insulin secretion [88]. Roux-en-Y gastric bypass increases the levels of GLP-1 [89], PYY [89], and oxyntomodulin post-prandially [97]. Moreover, Roux-en-Y gastric bypass decreases the levels of GIP, as it prevents the nutrient exposure to the duodenum and jejunum the sites where GIP is secreted in response to nutrients [98]. This novel triple combination, given by subcutaneous infusion, was studied in a recent single-blinded clinical study for 4 weeks, in 15 obese patients with diabetes or prediabetes [99]. Study outcomes showed that weight was reduced by 4.4 kg in the treatment group (−4.0% percentage change from baseline) compared to the placebo group, in which weight was reduced by 2.5 kg (−2% percentage change from baseline). The triple combination infusion also improved glycemia and reduced glucose variability. The infusion did not have any significant side effects, apart from mild erythema around the infusion site [92]. Introductory results for the above combination were provided by the study of Schmidt et al. The study showed that the co-infusion of GLP-1 and PYY3-36 reduces energy intake compared with placebo (−30.4 ± 6.5%) and enhanced the satiety effect following a meal, demonstrating a synergistic effect [100].

## 4. Discussion

Given the upward trend in the worldwide obesity prevalence and its numerous health and socioeconomic consequences, the need for combating obesity remains crucial. The existing treatment options include alterations in lifestyle and diet, as well as surgical and pharmacological approaches, with mechanisms that are not clearly understood and result in a variety of side effects. Besides the adverse effects, weight regain is a common homeostatic response after excessive weight loss and current anti-obesity medications show limited effectiveness in achieving long-term weight loss maintenance [101]. Consequently, the aim of treatment for overweight individuals is long-term weight reduction with maintenance, management of co-morbidities, improvement of the metabolic and cardiovascular profile and increase in the life expectancy [102]. A weight loss of 5% of body weight has been shown to be beneficial, while pharmacotherapy in combination with behavioral amendments achieving weight loss of 5–10% is considered satisfactory [96]. It is essential that drug therapy is used combined with a healthy diet, physical activity, and behavior alteration. The decision to initiate pharmacotherapy should be individualized and made after consideration of the risks and benefits when the individual’s BMI is over 30 kg/m^2^ or over 27 kg/m^2^ with an obesity-related disease [14]. Currently, bariatric surgery is the most effective therapeutic intervention for obesity but is approved only for the late stages of the disease. Weight reduction is mainly based on the hormonal alterations that lead to overall metabolic improvements. GLP-1 and PYY levels significantly increase after Roux-en-Y bypass due to increased deposition of large nutrient load to the distal gut. Levels of leptin, estrogen, and ghrelin decrease, while adiponectin, insulin, and cholecystokinin increase. GIP levels have inconsistent elevations or no change. This elevation in incretin hormones leads to increased satiety and improved insulin sensitivity. However, bariatric surgery is associated with various side effects due to hormonal imbalance, such as bone loss, vitamin deficiencies [103], and anatomical complications such as dumping syndrome, peritonitis, and superior mesenteric vein thrombosis [104]. Therefore, there is an urgency for less invasive treatments with a similar efficacy. Gut hormones, GLP-1, GIP, oxyntomodulin, and peptide YY lead to appetite suppression and reduced food intake, as well as increased energy expenditure [105]. Through their actions, the overall weight loss and metabolic profile of the patients may be improved. Their key role in weight regulation and metabolism makes them a promising potential target for pharmacotherapy. The development of combined anti-obesity medications (per os or injectable) that target more than one of these redundant molecular pathways, involved in weight regulation, as well as structures in CNS such as the habenula is a novel approach that may increase efficacy and improve tolerability. Combination pharmacotherapies may also achieve greater weight loss than monotherapies, via additive or synergistic effects, without causing major side effects. Data from recent clinical trials on GLP-1, GIP, glucagon receptor agonists, amylin analogues show positive results regarding weight loss (Table 2). Further clinical trials are required in order to strengthen whether these dual GLP-1/glucagon agonists are superior to GLP-1 analogues. Maintenance remains challenging and requires lifestyle and diet changes in combination with pharmacotherapy. However, the synergistic effect of combination agents is proven to be beneficial towards that goal. Moreover, triagonist compounds have already shown positive results in animal studies regarding nonalcoholic steatohepatitis and cholesterol-lowering effects.

## 5. Conclusions

In the field of obesity, combination therapies are currently under investigation. The existing anti-obesity medications have failed to provide sustainable weight loss over time. Moreover, various drugs that have reached advanced stages of clinical trials were withdrawn later due to safety issues. The approval of additional pharmacological therapies that aim to promote sustained weight loss in the obese population is mandatory. Intensive behavioral changes are also required in combination with pharmacotherapy, dietary changes, and exercise to produce the desired weight loss. The future of optimized anti-obesity pharmacotherapy is targeted towards weight maintenance and individualized combination therapy based on the metabolic profile of the person.

## Figures and Tables

**Table 1 pharmaceuticals-14-00869-t001:** Weight loss medications FDA-approved in the USA.

Drug	Brand Name	Average Weight Loss in Treatment vs. Control Group	% of Subjects Losing > 5% of Initial Body Weight
Liraglutide	Saxenda	−7.7 kg vs. −3.3 kg [15]	64.7% [15]
Orlistat	Xenical, Alli	−6.8% vs. −3.8% from baseline [16]	64% [16]
Semaglutide	Wegovy	−14.9% vs. −2.4% change in body weight from baseline [17]	86.4% [17]
Naltrexone/Bupropion sustained release	Contrave	−9.3 ± 0.4 kg vs. −5.1 ± 0.6 kg [18]	66.4% [18]
Phentermine/Topiramate extended release	Qsymia	−10.5% vs. −1.8% change in body weight from baseline [19]	50% [19]
Lorcaserin ^1^	Belviq	−5.8 ± 0.2 kg vs. −2.2 ± 0.1 kg [20]	47.5% [20]

^1^ In February of 2020 Lorcaserin was withdrawn from the US market.

**Table 2 pharmaceuticals-14-00869-t002:** Weight loss combined medications under investigation.

Drug	Average Weight Loss in Treatment Group versus Placebo Group	Side Effects	Clinical Trial Time Frame
GIP/GLP-1 Receptor Agonist [63]	−0.9 to −11.3 kg vs. −0.4 kg	Nausea, vomiting, diarrhea, decreased appetite	26 weeks
GLP-1/Glucagon Receptor Agonists [71]	−2.4 ± 0.4 kg vs. −0.5 ± 0.6 kg	Mild nausea, discomfort at injection site	4 weeks
SGLT2/GLP-1 Receptor Agonists [87]	−3.55 (−4.12 to −2.99) vs. −1.56 (−2.13 to −0.98) or −2.22 (−2.78 to −1.66)	Diarrhea, nausea, injection site nodules, urinary tract infections	28 weeks
GLP-1 Receptor Agonist/Amylin Analogue [92]	−15.9 (*SE* 1.40) vs. −7.8 (*SE* 2.2)	Nausea (most common), vomiting, dyspepsia, decreased appetite	20 weeks
GLP-1/GIP/Glucagon Receptor Agonists [93] ^1^	−26.6% vs. −15.7%	-	20 days

^1^ Animal rodent/mice studies. *SE*: standard error.

## Data Availability

Data sharing not applicable.

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
