# Peer review of "New Incretin Combination Treatments under Investigation in Obesity and Metabolism: A Systematic Review"

_pharmaceuticals, 2021, doi:10.3390/ph14090869_

Round 1
Reviewer 1 Report
In this review, Kakouri et al., offer an interesting review of studies, literature focused on the use of incretins, alone or in combination, for the treatment of obesity and associated metabolic complications.
Altogether, I think this work would be a useful addition to the scientific literature. Nevertheless, there is, in my opinion, some space for improvement.
While a review is a helpful tool, it is best when it can almost stand on its own. Here, unfortunately, there is some work to do, in my opinion, in order to improve the review so that the reader does not feel compelled to go to the cited study(ies) to understand what the written statement in the review really means. It sometimes lacks of a bit more details. Some statements feel too general, without real direction.
For example: lines 297-299: “Moreover, GLP-1 receptor agonists and SGLT2 inhibitors lower visceral fat, and the decrease is disproportionate to the decrease in total body fat [73].” “Disproportionate” -> how? Regarding the subject, is it something rather positive? Is there only the need to target the visceral fat?
Although a review aims at summarizing findings, I think findings could appear a bit clearer and sometimes discussed just a bit more. Yet, I’d like to mention that it is not a constant in the manuscript: some paragraphs are already a very useful and clear summary of the literature.
Overall, with some little improvement here and there, I believe this work would definitely find its audience to which it would provide interesting information and insight.
I also listed some aspects that could be directly improved below:
- Lines 47-48: “The pathophysiology of the disease is complex, and multiple genetic, metabolic, behavioral, and environmental factors are involved.”.
- It might be preferable to avoid multiplying commas. It does not read too well.
- Lines 197-199: “The actions of GIP and GLP-1 are mediated by the engagement of G- protein-coupled receptors expressed on pancreatic α- and β-cells and in peripheral tissues, including the gastrointestinal tract, kidneys, central nervous system, heart, and lungs.”.
- I would recommend to remove the term “peripheral” as CNS is commonly not considered a peripheral tissue.
- Lines 216-226: please associate references.
- Line 240: “26-week”, not “26-weekS”
- Lines 246-247: “14-71% of subjects treated with tirzepatide achieved at least 5% weight reduction [59].”
- To better understand, I had to go back to the cited study because, although the authors in this review mention different BW loss values for different tirzepatide doses, it is not super evident to grasp here, in my opinion at least.
- Lines 269: “overweight” not overweightED”.
- Lines 268-270: I would suggest some rewriting here. It does not read really well.
- Lines 274-276: Same as above. I would suggest some rewriting here as well.
- Regarding GLP-1/Glucagon agonists, I would definitely discuss why there is an ongoing search for synthetized molecules when oxyntomodulin naturally occurs. Higher latency before clearance ? etc.
- Line 284: “49-day”, not “49-dayS”
- Line 285: maybe replace “>” with ”over”.
- Lines 291-293: Could benefit from rewriting. Especially the part of the sentence with GLUTs that is confusing.
- Lines 297-299: Please give more precisions.
- Line 302: “placebo and licogliflozin”: don’t you mean “or”?
- Line 313: “28-week”, not “28-weekS”
- Lines 374-375: Writing? Two sentences?
- Lines 378-380: Writing?
- Line 391: missing comma after “understood”
- Line 397: “increase IN”
- Line 427: “Maintenance”.
Author Response
Please see the attachment
We thank the reviewer for the important and well-researched feedback and comments regarding our manuscript.

Reviewer 2 Report
The paper entitled “New incretin combination treatments under investigation in obesity and metabolism” includes potentially relevant data for the pharmacology of the metabolic and inflammatory disorders. The Authors of the manuscript conclude suggesting that anti-obesity agents target more than one of the molecular metabolic pathways involved in energy substrate turnover regulation.
However, I have a few remarks:
- The first sentence of Introduction section needs to be remodeled, “Obesity is a chronic metabolic disease, caused by increased body fat accumulation and is correlated with negative effects on people’s health”, “increased body fat accumulation” rather seems to be the symptom but not the cause of metabolic disorder in question;
- It is absolutely necessary – at least briefly – discuss the most important interactions of the described active substances;
- It is highly recommended – at least shortly – list the most crucial contraindications of the described active pharmaceutical ingredients;
- Not only in the Abstract section but also in the Introduction section I did not find a clearly defined purpose/aim of the paper;
- Due to the lack of a clearly defined aim of the work, I am not able to determine the accuracy of the conclusions proposed by the authors;
- There are numerous Pharmaceutical / Pharmacological sources of information, which include, for example: (DrugBank, Embase, International Pharmaceutical Abstracts, Biosis Citation Index, SciFinder, Micromedex, Lexi-Comp, Facts and Comparisons E-Answers, Clinical Pharmacology (Clinical Key interface), Health Sciences Mobile Resources Guide); for unknown reasons, the Authors used PubMed and The Clinical-Trials.gov only. Explanation is highly needed.
Author Response
Please see the attachment
We thank the reviewer for the meaningful comments.

Round 2
Reviewer 1 Report
The authors have generally addressed the comments of both reviewers and the quality of the manuscript has increased because of it. The manuscript appears, in its last version, suitable for publication.
Reviewer 2 Report
The Authors of the manuscript responded to the comments of the reviewer.
The Authors introduced appropriate corrections to the manuscript.
The manuscript of the publication - in its current form - is suitable for publication, without any corrections.